# Analysis of Depigmenting Substances of Interest (Hydroquinone, Kojic Acid, and Clobetasol Propionate) Contained in Lightening Cosmetic Products Marketed in Burkina Faso

Boumbéwendin Gérard Josias Yaméogo [1,2,*], Lydiane Sandra B. A. Ilboudo [1], Nomtondo Amina Ouédraogo [3], Mohamed Belem [2], Ouéogo Nikiema [2], Bertrand W. Goumbri [2], Bavouma Charles Sombié [1], Hermine Zimé-Diawara [1], Elie Kabré [2] and Rasmané Semdé [1]

[1] Drug Development Laboratory (LADME), Center for Training, Research and Expertise in Medicine Sciences (CEA-CFOREM), Doctoral School of Sciences and Health (ED2S), University Joseph KI-ZERBO, Ouagadougou 03 BP 7021, Burkina Faso; dianacarly16@gmail.com (L.S.B.A.I.); charlsombie@yahoo.fr (B.C.S.); zimegani@yahoo.fr (H.Z.-D.); rsemde@yahoo.fr (R.S.)

[2] National Agency for Health Safety of the Environment, Food, Work and Health Products (ANSSEAT), Ministry of Health, Ouagadougou 09 BP 24, Burkina Faso; rabankibelem70@gmail.com (M.B.); oueogonikiema@gmail.com (O.N.); bertrand.bribera@gmail.com (B.W.G.); elie.kabre@gmail.com (E.K.)

[3] Service of Dermatology-Venereology, University Hospital Center Yalgado Ouédraogo, University Joseph KI-ZERBO, Ouagadougou 03 BP 7021, Burkina Faso; nomtondo2000@yahoo.fr

[*] Correspondence: josiasbyg@yahoo.fr; Tel.: +00226-70-06-12-92

**Abstract:** The practice of voluntary depigmentation is still prevalent in Africa, with a wide range of lightening cosmetics used. Our objective was to research and quantify three regulated and/or prohibited depigmenting ingredients present in lightening cosmetics sold in Ouagadougou. Twenty-nine lightening cosmetic samples were collected from vendors and HPLC analysis was subsequently conducted to identify and measure the concentrations of hydroquinone, clobetasol propionate, and kojic acid. The presence of hydroquinone was indicated on the label of 13.79% of the products, while 51.72% contained it after analysis. Furthermore, none of the products mentioned a concentration of hydroquinone exceeding 2.00%, even though 27.58% of them contained high concentrations. For clobetasol propionate, its presence was stated on the labels of 13.79% of the products, while 31.03% contained it. One sample had a clobetasol content exceeding 0.05%, although none mentioned a concentration higher than this value. Finally, while 24.13% of the samples claimed to contain kojic acid, only 17.24% did. We also observed that 41.38% of the samples contained combinations of two depigmenting ingredients investigated, with a predominance of the hydroquinone + clobetasol propionate (27.38%). These results demonstrate that manufacturers' declarations regarding the compositions of active ingredients in lightening cosmetics can sometimes be deceptive.

**Keywords:** lightening cosmetic; hydroquinone; clobetasol propionate; kojic acid; HPLC analysis



## 1. Introduction

Voluntary cosmetic depigmentation is defined as all practices aimed at lightening the skin through the cosmetic use of products with clearly established depigmenting properties [1]. It is mainly practiced by women in Africa [2–4]. Its prevalence among women in Nigeria was estimated at 40.09% in 2021 [4]. In Burkina Faso, Andonaba et al. reported a prevalence of 49.2% in the city of Bobo-Dioulasso in 2016 [5]. The prevalence in the city of Ouagadougou is lower and had been estimated in 2005 at 39.5% [6]. The harmful effects of this practice on health are numerous [6–9], and in some Black African countries, it constitutes a real public health problem [2,6,9–11].

The depigmenting agents usually used are of natural or synthetic origin, and are most often used in combination. They most often consist of a strong class of dermocorticoids (for

example, clobetasol propionate), hydroquinone in variable concentrations ranging from 2.00 to 8.00%, or keratolytics (salicylated vaseline with concentrations of up to 50%). There are also homemade preparations containing mercury salts, soda-based soaps, and oxidizing mixtures (based on bleach, hydrogen peroxide, peroxides, or perchlorates, etc.) [2,9,12,13]. Natural substances used in this practice include kojic acid, alpha arbutin, vitamin C, azelaic acid, retinoids, glutathione, and alpha-hydroxy acids (AHAs) [12,14–16]. Andonaba et al. showed that, based on label statements, 81.6% of the lightening cosmetics marketed in the city of Bobo-Dioulasso in 2017 contained hydroquinone. The remaining products contained various mixtures (11.12%), EDTA (8.33%), kojic acid (4.86%), and unknown substances (14.58%). They also revealed that 98.96% of products did not bear any indication of their origins [5]. This raises the question of whether the claims made on the labels of these products are accurate and do not relate to misleading commercial practices.

It is with this in mind that we conducted this study, the aim of which was to verify the concordance between the active ingredients (presence and rate of incorporation) mentioned on the label and the actual content of these lightening cosmetic products. After assessing the physical and regulatory characteristics of a sample of lightening cosmetics collected in the city of Ouagadougou, we carried out an analytical screening to detect the presence of hydroquinone (and its ethyl, methyl, and benzyl ethers), clobetasol propionate, and kojic acid in these products, and to quantify these three active ingredients. These three depigmenting substances were targeted because they are the most commonly used in lightening cosmetic products at very high percentages, implying numerous side effects [13,17,18].

## 2. Materials and Methods

### 2.1. Material

Kojic acid (99.0%) and hydroquinone monoethyl ether (99.0%) were purchased from Sigma-Aldrich (Taufkirchen, Germany). Beclomethasone (99.9%) was from the European Pharmacopoeia (Ph. Eur) and clobetasol propionate (99.9%) was from the United States Pharmacopeia (USP, Rockville, MD, USA). Hydroquinone monomethyl ether (98.0%) and hydroquinone monobenzyl (99.0%) were supplied by Fluka (Buchs, Germany) and hydroquinone (99.0%) was supplied by Panreac (Castellar del Vallès, Barcelona, Spain). Hydrochloric acid (37.0%) and orthophosphoric acid (85.0%) were from VWR Chemicals (Fontenay sous bois, France). Methanol and acetonitrile were HPLC grade and were obtained from Carlo Erba Reagents (Val-de-Reuil, France). Tetrahydrofuran for analysis was obtained from Scharlau (Sentmenat, Spain).

### 2.2. Methods

2.2.1. Sampling

Sample collection sites were the main stores selling cosmetics in the 12 districts of Ouagadougou. These stores were identified based on information obtained from hair salons, beauty salons, and users. To collect the samples, one store was selected in each district, for a total of 12 stores. These 12 stores were chosen based on their reputation for selling lightening products and the diversity of the product range on offer.

We went to each of the 12 selected boutiques to ask the managers for advice, expressing our desire to lighten our skin. After discussions with the managers, we bought two or three products in each boutique, based on the advice we had received and taking care not to use any brand of product we had already bought in previous boutiques. During all stages of the study, the samples collected were kept under the storage conditions specified by the manufacturer or, if necessary, at 20 °C in an air-conditioned room.

2.2.2. Evaluation of Sample Characteristics

The physical form and labeling characteristics of the samples collected were assessed against the regulatory requirements of the WAEMU (West African Economic and Monetary Union) [19]. The information required on the label and/or packaging of cosmetic products

was documented. This information includes the brand name, claimed properties, depigmenting substances mentioned on the packaging, the identity, address, and origin of the manufacturer, INCI list (International Nomenclature of Cosmetic Ingredients), capacity, batch number, date of manufacture, expiration date, role, and terms of use of the product.

### 2.2.3. Identification and Assay of Hydroquinone and Its Ether Derivatives

The method for detection and assay of hydroquinone and its ether derivatives was slightly adapted from the standard BS EN 16956:2017 [20]. A sample of each cosmetic product was accurately weighed (Metler Toledo XPE206DR balance) and extracted with water/methanol solvent (50:50 $v/v$) at 60 °C. The filtrate was analyzed using high-performance liquid chromatography (Agilent 1260 infinity HPLC system coupled to a diode array detector (DAD)). The chromatographic conditions used were as follows: wavelength 295 nm, mobile phase water/tetrahydrofuran (55:45 $v/v$), elution flow rate 1 mL/min, injection volume 10 μL, column temperature 30 °C, ODS-C18 column (250 mm × 4.6 mm × 5 μm).

The reference solution was prepared extemporaneously as indicated in BS EN 16956:2017 [20].

The analytical method was verified as described in the ICH guide to reproducibility and repeatability parameters [21].

### 2.2.4. Identification and Assay of Kojic Acid

Kojic acid was extracted from 2.0 g ± 0.1 g of sample with a 0.1 N hydrochloric acid solution and then filtered. The kojic acid reference solution (200 μg/mL) was prepared by dissolving 1.0 mg kojic acid standard in a 50.0 mL mobile phase [22]. The reference solution and extracts were analyzed using HPLC-DAD under the following chromatographic conditions: wavelength 214 nm, mobile phase methanol/distilled water/orthophosphoric acid (250 mL + 750 mL + 3.0 mL), flow rate 0.8 mL/min, injection volume 5.0 μL, column temperature 40 °C, XDB-C8 column (160 mm × 4.6 mm × 5 μm). The reproducibility and repeatability of the method were also checked according to ICH guidelines [21].

### 2.2.5. Identification and Assay of Clobetasol Propionate

The identification and assay of clobetasol propionate from the sample was carried out according to USP monograph. For the extraction of clobetasol propionate and reference solution preparation, methanol was used as solvent [23].

The reference solution of clobetasol propionate was prepared as follows: First, a 0.2 mg/mL internal standard solution of beclomethasone dipropionate in methanol was prepared. To 10 mL of this internal standard, 1 mg of USP clobetasol propionate reference substance was added, then the volume was made up to 25 mL with methanol [23].

For the extraction of clobetasol propionate, 10 mL of internal standard and 15 mL of methanol were added to the test sample (2 ± 0.1 g), followed by vigorous shaking, centrifugation at 3500 rpm for 10 min, and filtration [20].

The reference solution and extracts were analyzed using HPLC-DAD at a wavelength of 240 nm, using a mobile phase consisting of methanol/phosphate buffer pH 5.5/acetonitrile (10:47.5:42.5 $v/v/v$). The elution flow rate was 1 mL/min and the injection volume was 10 μL. An L1 column (150 mm × 4.6 mm × 5 μm) was used at a temperature of 30 °C. The system suitability was checked before the start of the analyses, according the requirements of the USP [23]. The limit of detection (LOD) and limit of quantification (LOQ) were 0.45 and 1.4 μg/mL, respectively.

### 2.2.6. Data Validation and Statistical Analysis

All the equipment used was qualified and suitable for its intended use.

All measurements were carried out in triplicate and results expressed as the average percentage of the three analyses.

Retention times used for the identification of hydroquinone and its derivatives were 3.7 min for hydroquinone, 4.2 min for monomethyl ether, 4.5 min for monoethyl ether, and

5.9 min for monobenzyl ether. Retention times for kojic acid and clobetasol propionate were 2.4 min and 7.4 min, respectively.

## 3. Results

### 3.1. Characteristics of Samples Collected

3.1.1. General Characteristics

A total of 29 samples of lightening products from different brands were collected. Information on the samples collected is presented in Table 1 below.

**Table 1.** Characteristics of lightening cosmetics samples collected.

| Product Code | Claimed Properties | Depigmenting Substances Mentioned on the Packaging |
| --- | --- | --- |
| S1 | Treatment and lightening concentrate | Fruit acids, Vitamin C, Collagen |
| S2 | Skin lightener | Glutathione, Carrot oil |
| S3 | Lightening cream | None |
| S4 | Anti-spot whitener | Kojic acid, Snail slime |
| S5 | Lightening, clear, and beautiful without blemishes | Carrots |
| S6 | Bleaching | None |
| S7 | Anti-spot cream | Clobetasol propionate < 0.05%, Glutathione |
| S8 | 100% lightening collagen oil | Glutathione 5%, Glycolic acid 48%, Lactic acid 12%, Salicylic acid 3%, Kojic acid 9% |
| S9 | Clarifying beauty treatment serum | Carrot oil, Salicylic acid, Snail slime |
| S10 | Clarifying milk | Carrot oil, Hydroquinone < 2%, Arbutin |
| S11 | Whitener | Arbutin |
| S12 | Vitamin C body serum | Vitamin C |
| S13 | Lightening beauty milk | Vitamin C, Kojic acid, Carrot oil |
| S14 | Super lightener with fruit acids | Kojic acid, Glutathione |
| S15 | Concentrated clarifying milk | Fruit acids, Glutathione, Carrot oil, Ascorbic acid |
| S16 | Specific lightening toner: elbows, hands, knees | None |
| S17 | Lightening oil | Hydroquinone, Carrot oil |
| S18 | Super lightening with fruit acids, Anti-spots | Kojic acid |
| S19 | Blemish-free, clear complexion | Hydroquinone < 2%, Ascorbic acid |
| S20 | Extra-concentrated unifying serum | Fruit acids |
| S21 | Lightening treatment oil | Vitamin E |
| S22 | Extra-strong treating and clarifying milk | AHA |
| S23 | Lightening care oil | Kojic acid, Salicylic acid |
| S24 | Whitening milk | Fruit acids, collagen |
| S25 | Fast-acting | Clobetasol propionate < 0.05% |
| S26 | Skin repair gel | Clobetasol propionate < 0.05% |
| S27 | Ultra-whitening | Clobetasol propionate < 0.05% |
| S28 | Clarifying beauty milk | Carrot oil, Hydroquinone < 2 |
| S29 | Treatment and whitening of acne spots | Kojic acid, Glutathione |

3.1.2. Product Forms

Figure 1 depicts the distribution of the collected samples based on their dosage forms. Oils/serums (44.59%) and creams (37.93%) were the two most commonly used dosage forms of skin-lightening cosmetics.

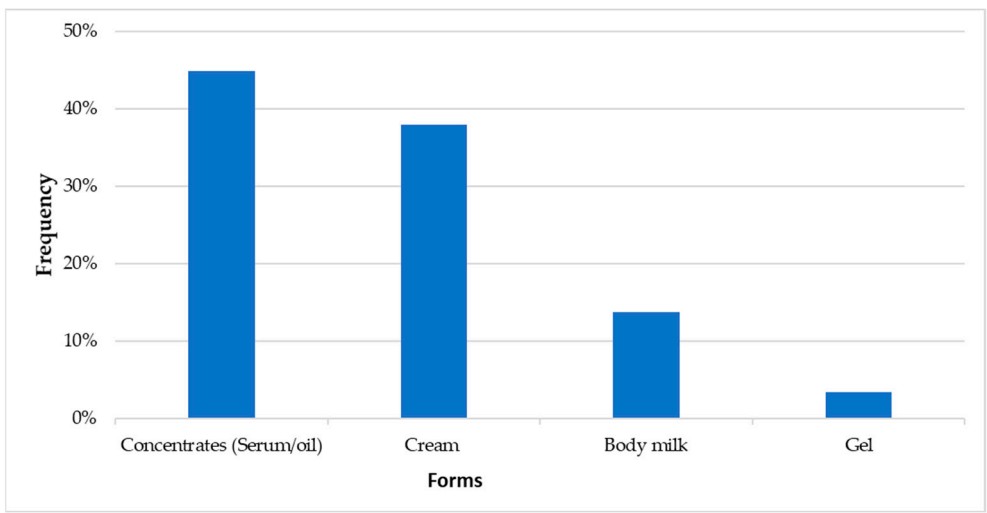

**Figure 1.** Distribution of skin-lightening cosmetic samples according to their dosage forms.

### 3.1.3. Manufacturer's Origin

Table 2 presents the distribution of collected samples according to their origins. It should be noted that the country of origin of the manufacturer was not specified in 20.69% of the collected samples. Apart from these products, Côte d'Ivoire and Togo were the two main suppliers of the collected skin-lightening cosmetics.

**Table 2.** Distribution of collected samples of skin-lightening cosmetics according to the manufacturer's origin.

| Manufacturer's Country of Origin | Total (*n* = 29) | Percentage (%) |
|---|---|---|
| Cameroon | 2 | 6.90 |
| Egypt | 1 | 3.45 |
| France | 1 | 3.45 |
| India | 1 | 3.45 |
| Italy | 2 | 6.90 |
| United Kingdom | 1 | 3.45 |
| The Philippines | 1 | 3.45 |
| Côte d'Ivoire | 6 | 20.69 |
| Senegal | 1 | 3.45 |
| Thailand | 1 | 3.45 |
| Togo | 5 | 17.24 |
| USA | 1 | 3.45 |
| Not stated | 6 | 20.69 |

### 3.2. Evaluation of Product Labeling

#### 3.2.1. Compliance with Labeling Rules

Table 3 displays the distribution of collected samples according to the various mandatory labeling statements found on the packaging. It was observed that the least-adhered-to statements were the lot number and the manufacturing date.

**Table 3.** Compliance with labeling statements of collected skin-lightening cosmetics.

| Mandatory Information | Number of Samples Bearing the Statement | Conformity Rate (%) |
|---|---|---|
| INCI List | 23 | 79.31 |
| Manufacturer's identity | 22 | 75.86 |
| Manufacturer's address | 20 | 68.96 |
| Manufacturer's country of origin | 23 | 79.31 |
| Capacity | 24 | 82.75 |
| Batch number | 7 | 24.13 |
| Manufacturing date | 9 | 31.04 |
| Expiration date | 18 | 62.06 |
| Role of product | 25 | 86.20 |
| Product use modalities | 17 | 58.62 |

INCI: International Nomenclature of Cosmetic Ingredients.

### 3.2.2. Mentions of the Presence of Depigmenting Substances on Labels

Based on label statements, 82.76% of the collected skin-lightening cosmetics explicitly indicated the presence of depigmenting substances. Among these products, 31.03% claimed the presence of a single depigmenting substance, while 51.72% reported combinations of multiple depigmenting substances. Figure 2 illustrates the distribution of samples according to the number of depigmenting substances mentioned on the packaging.

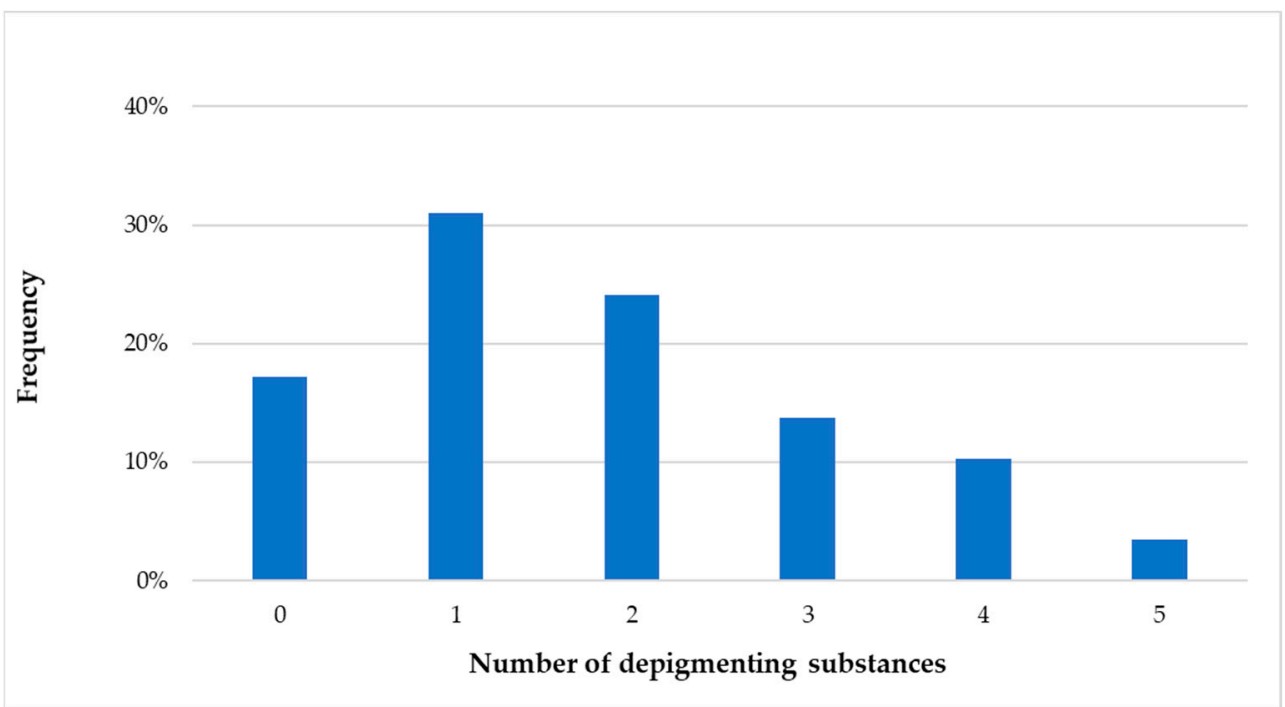

**Figure 2.** Distribution of the samples based on the number of depigmenting substances declared on the packaging.

Table 4 shows the distribution of collected samples that claimed the presence of the three depigmenting substances of interest (hydroquinone, kojic acid, and/or clobetasol propionate) on their labels.

**Table 4.** Distribution of collected samples claiming the presence of hydroquinone, kojic acid, and/or clobetasol propionate.

| Depigmenting Substances Clamed | Number of Products | Percentage (%) |
|---|:---:|:---:|
| Mention of the presence of hydroquinone | 4 | 13.79 |
|   -     Without specified concentration | 0 | 0.00 |
|   -     Specified concentration < 2% | 4 | 13.79 |
| Mention of the presence of clobetasol propionate at a concentration < 0.05% | 4 | 13.79 |
| Mention of the presence of kojic acid | 7 | 24.12 |
|   -     Without specified concentration | 6 | 20.68 |
|   -     With specified concentration | 1 | 3.44 |

As can be observed, kojic acid was the most commonly mentioned depigmenting substance (24.12%) on the labels of the collected samples.

### 3.3. Results of Screening Tests and Assay for Depigmenting Substances

The results of the screening analyses and assay showed that the presence of hydroquinone was mentioned in four (4) products (13.79%), while fifteen (15) products (51.72%) contained it. Furthermore, the presence of hydroquinone at concentrations greater than 2% was not mentioned on any product, but eight (8) products (27.58%) contained it at higher concentrations. Clobetasol propionate was declared in four (4) products (13.79%), while nine samples (31.03%) contained it. Additionally, one (1) sample had a clobetasol propionate content exceeding 0.05%, even though no sample mentioned a concentration higher than this value. Lastly, while seven (7) samples, or 24.13%, claimed the presence of kojic acid, only five (5) products, or 17.24%, actually contained it.

Figure 3 provides a comparison of the proportions of products claiming the presence of each of the three sought-after depigmenting substances and those that contained them.

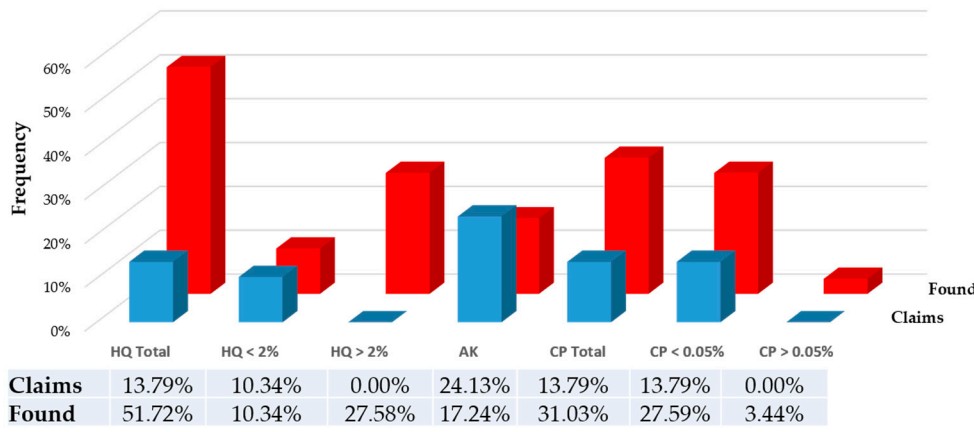

| | HQ Total | HQ < 2% | HQ > 2% | AK | CP Total | CP < 0.05% | CP > 0.05% |
|---|---|---|---|---|---|---|---|
| **Claims** | 13.79% | 10.34% | 0.00% | 24.13% | 13.79% | 13.79% | 0.00% |
| **Found** | 51.72% | 10.34% | 27.58% | 17.24% | 31.03% | 27.59% | 3.44% |

**Figure 3.** Distribution of the samples based on claimed and experimentally found depigmenting substances.

## 4. Discussion

### 4.1. Characteristics of Collected Samples

Twenty-nine (29) samples of cosmetic products from different brands claiming 'lightening', 'clarifying', or 'whitening' properties were collected. These products were in four different dosage forms: concentrates oil/serum (44.59%), cream (37.93%), lotion (13.79%), and gel (3.45%). Nyiragasigwa noted in 2021 that there was a concurrent use of soap and cream in 61.5% of cases among Black populations in Belgium [24].

The country of origin of the manufacturer was not specified in 20.69% of the collected samples. The products collected were manufactured in the neighboring countries of

Burkina Faso, such as Côte d'Ivoire (20.69%) and Togo (17.24%), accounting for 37.93% of the products. Products manufactured in other African countries (Cameroon, Egypt, Senegal) represented 13.79%. The remaining 27.58% were imported from Europe (France, Italy, England), India, the Philippines, Thailand, and the United States of America (USA). No cosmetic product from the collected brands was locally manufactured in Burkina Faso. Tra also mentioned in her study that the basic products used for skin depigmentation in Burkina Faso were all imported. However, mixtures of these imported brand products were usually prepared locally by sellers for the specific needs of each customer [25].

### 4.2. Compliance with Labeling Rules

Compliance with labeling rules was assessed following the WAEMU guidelines [19]. The five most frequently indicated labeling statements on the containers and/or packaging of collected products were the product's purpose (86.20%), the content (indicated using weight or volume) (82.75%), country of origin (79.31%), the list of ingredients or INCI list (79.31%), and the manufacturer's identity (75.86%).

The INCI list is a mandatory nomenclature for cosmetic products. Manufacturers are not obligated to indicate the concentration of each ingredient due to "trade secrets", but they must list them in descending order of their weight if they are dosed at more than 1.00%. Below 1.00%, the manufacturer can list them in any order on the packaging [19].

The mention of the product's indication, which appeared on 86.20% of the samples, is a safety requirement of the WAEMU guidelines. It aims to protect human health and provide clear information to consumers regarding the product's use and application [19].

Out of the 23 products (79.31%) that mentioned the country of origin of the manufacturer, 22 (75.86%) included the manufacturer's identity, and 20 (68.96%) clearly stated the manufacturer's address, following WAEMU guidelines [19].

The expiry date or expiration date was mentioned on 62.06% of the products. This date is defined as the date until which the product, when stored under appropriate conditions, continues to fulfill its function. It is indicated on products mainly as "Best before (date)" or as the duration of use after opening the bottle, expressed in months or years. Indeed, WAEMU guidelines specify that for cosmetic products with a minimum durability exceeding thirty months, mentioning the expiry date is not mandatory. These products can only indicate the allowed duration of use after opening without harm to the consumer [19].

The mention of the manufacturing lot number was the least indicated statement on the samples, appearing on only 24.13% of them. The manufacturing lot number, indicated as numbers and/or letters, serves to identify and track a set of identical products that share certain production characteristics (time and date of production, identification code, etc.). This lot number ensures product traceability and its historical and contextual data [19].

### 4.3. Mentions of the Presence of Depigmenting Substances on Labels

In cosmetics, the active ingredient is the guarantor of the product's properties, and its mention on the label and/or packaging is mandatory [26].

The evaluation of the labels and/or packaging of the collected products shows that 82.76% explicitly indicated the presence of one (31.03% of cases) or multiple depigmenting ingredients (51.72%). The most frequently mentioned depigmenting molecules were, respectively, kojic acid, hydroquinone, clobetasol propionate, AHAs, niacinamide, and retinol.

Kojic acid, or 5-hydroxy-2-(hydroxymethyl)-4-pyrone, is a natural substance produced by several species of fungi, especially Aspergillus oryzae [27]. Its presence was mentioned on the labels of 24.12% of the collected products. Andonaba et al. [5] found in 2019 that only 4.86% of skin-lightening cosmetics marketed in the city of Bobo-Dioulasso in Burkina Faso mentioned the presence of hydroquinone. Indeed, since 2001 there has been a ban on the use of hydroquinone in cosmetic products in Europe [28]. Due to its numerous adverse effects, the use of kojic acid as a skin-lightening or depigmenting agent has become widespread. The natural origin of kojic acid and its fewer adverse effects have been highlighted, and all

manufacturers would like to display it on their packaging to attract customers. However, it is important to note that kojic acid has a strong allergenic potential with a relatively high frequency of contact dermatitis and erythema when improperly used on the skin. It also possesses antioxidant, antibacterial, and antifungal properties [12,29].

Hydroquinone and its derivatives were mentioned on the label and/or packaging of 13.79% of the collected samples. Tra [25] also noted in 2019 a low rate of 15.0%, compared to a previous study's data from 2017 in Bobo-Dioulasso, Burkina Faso, in which 81.6% mentioned the presence of hydroquinone [5]. Hydroquinone and its derivatives have been a reference for depigmenting agents for many years. However, our results show that they are being used less and less now. This could be explained by the presence of new depigmenting molecules on the market such as kojic acid. It is also possible that manufacturers avoid declaring the presence of hydroquinone to avoid attracting consumers' attention. These manufacturers may continue to incorporate it into their preparations without mentioning it on the label and/or packaging because it is inexpensive and effective. In this latter case, it constitutes deceptive claims. Indeed, hydroquinone and its derivatives are powerful melanogenesis inhibitors, and highly effective in skin whitening. However, they have numerous adverse effects that have led to their prohibition in cosmetic products in the European Union [28]. They are highly cytotoxic and are involved in the development of ochronosis after prolonged use [30]. Despite the evidence of their toxicity, the incorporation of hydroquinone in cosmetics remains allowed in WAEMU countries with an exemption dose of 2% [19].

Clobetasol propionate was declared at a concentration <0.05% in 13.79% of the samples. It is a dermocorticoid, a topical steroid anti-inflammatory used locally. As depigmentation is a side effect, clobetasol propionate is used as a whitening agent in illegally sold cosmetic preparations. Indeed, the incorporation of dermocorticoids in cosmetic products is strictly prohibited in WAEMU countries, regardless of their concentration [19]. They are only allowed in dermatological medications at the recommended therapeutic dose of 0.05%.

It is important to emphasize that some products mentioned the presence of several active ingredients, often with up to five depigmenting molecules. Combinations of two, three, four, or five depigmenting molecules represented 24.14%, 13.79%, 10.34%, and 3.45% of the samples, respectively.

Finally, mentions of certain adjuvants were noted on the labels and/or packaging of the collected products. These adjuvants aim to facilitate the cutaneous penetration of active ingredients. They include active ingredients such as AHAs and retinoic acid, and actual adjuvants such as propylene glycol [31].

### 4.4. Screening and Assay of Hydroquinone, Kojic Acid, and Clobetasol Propionate

Analytical screening was conducted on the 29 collected lightening cosmetic products to detect the presence of hydroquinone and its derivatives, clobetasol propionate, and kojic acid. These three molecules were the most frequently mentioned on the product labels. They are among the most toxic substances [17], and their use is subject to strict regulations [19,28]. The results showed that 24 products, or 82.76% of the samples, contained one or more of these three depigmenting molecules, either alone or in combination, while only 51.72% mentioned their presence. Subsequent dosage tests for these three molecules were performed on the products where their presence was detected.

The analytical results showed that 51.72% of the samples contained hydroquinone, while only 13.79% declared its presence. We also found that 27.59% of the products contained hydroquinone concentrations exceeding 2%, while no product had indicated such high concentrations. It should be noted that among the products that expressly claimed to contain hydroquinone at a concentration below 2%, two (2) of them contained it but at a higher concentration. In 2019, Tra [25] found that 44.12% of samples of cosmetic products collected in Burkina Faso and Côte d'Ivoire contained hydroquinone at concentrations below 2%, even though its presence was not mentioned on the labeling. Furthermore, in 2014, a study conducted in West Africa and Canada showed that 38.00% of samples had

hydroquinone concentrations exceeding the standards [13]. On the other hand, Siyaka et al. in 2016 collected 20 samples of lightening creams in Nigeria and found that all of them contained hydroquinone at percentages ranging from 0.07 to 4.00% [32]. The high levels of hydroquinone (exceeding 2%) detected in 27.59% of the samples in our study expose users to the risk of exogenous ochronosis and other health problems [17,30].

The analyses also revealed that 31.03% of the samples contained clobetasol propionate, with one sample having a concentration of the active ingredient exceeding 0.05%. However, the presence of this active ingredient was only declared in 13.79% of the samples, and none indicated doses exceeding 0.05%. There was a slight decrease in the use of this ingredient in our study compared to that of Gbetoh and Amyot [13] in 2014, who detected the presence of clobetasol propionate in 39.00% of the samples. Our results show that clobetasol propionate is incorporated in small quantities into lightening preparations and is generally combined with another depigmenting agent to have a synergistic effect. However, clobetasol propionate is a potent local corticosteroid that should only be dispensed through medical prescription. Preparations containing it should not be available in cosmetic product shops. The inappropriate use of this potent corticosteroid exposes individuals to increased risks of serious adverse effects such as the spread and worsening of untreated infections, irreversible skin thinning, dermatitis, acne, and hypertrichosis [9].

HPLC analyses also revealed that, out of the seven samples (24.13%) claiming the presence of kojic acid, only five products (17.24%) contained it. The incorporation levels of kojic acid found were still low and did not reach 1%, which is the allowed concentration in Europe [28]. These results lead us to believe that manufacturers often declare the presence of kojic acid, which is popular among consumers, but may not use it because its cost is very high compared to that of hydroquinone and clobetasol propionate. However, in 2018, Verdoni et al. found cases of lightening cosmetic products collected in Paris and Benin that contained kojic acid even though it was not listed on their labels [33]. It is important to note that WAEMU community regulations are silent on the use of kojic acid in cosmetic products.

Based on our laboratory analysis results, we observed that 12 cosmetic products in our sample, representing 41.38% of the total, contained combinations of two of the three molecules sought after. The combinations found were hydroquinone + clobetasol propionate (8 products, or 27.59% of the samples), hydroquinone + kojic acid (3 products, or 10.34% of the samples), and kojic acid + clobetasol propionate (1 product, or 3.45% of the samples). Verdoni et al. also found in their study that three products out of seven contained both hydroquinone and kojic acid [33].

The analytical data obtained show that 58.62% of the products analyzed were non-compliant due to the presence of prohibited active ingredients or because they exceeded the maximum content established through current regulations. These active ingredients were hidden from the user in some cosmetic products and sometimes their concentrations indicated on the packaging were deliberately falsified.

These products can be harmful to human health. Indeed, hydroquinone is a cytotoxic active ingredient and is involved in the occurrence of exogenous ochronosis [17,30]. Clobetasol propionate is a powerful topical corticosteroid responsible for many side effects, such as cutaneous immunosuppression, which is dose-dependent, irreversible thinning of the skin, dermatitis, induced acne, irregular and persistent hypochromia, rebound hyperpigmentation, stretch marks, and infectious complications. Regular and heavy absorption of clobetasol through the skin can cause adrenal suppression or even Cushing's syndrome [9]. Kojic acid has a high sensitization potential with a relatively high frequency of contact dermatitis and erythema. A combination of these substances in the same formulation undoubtedly makes it possible to obtain a synergistic effect with significant depigmenting activity, but will also lead to an increase or aggravation of their side effects.

Regulatory measures must be undertaken at the national level, in order to control the market of depigmenting products and protect the health of consumers.

## 5. Conclusions

Our study allowed us to verify the conformity of claims related to hydroquinone, clobetasol propionate, and kojic acid in cosmetic products marketed in the city of Ouagadougou. The analytical data show that the declarations on the labels of these products are often misleading. These results should be used to establish stricter regulations on cosmetic products in the WAEMU region and to raise awareness among stakeholders.

An in vitro analysis of these products on specific cell lines is underway with the aim to evaluate their potential damage action on the skin tissue.

Finally, the development of an analytical method that allows for the simultaneous detection and quantification of these three molecules in the same sample will enable better tracking of these three depigmenting substances in cosmetic products sold at the community level.

**Author Contributions:** Conceptualization, B.G.J.Y. and N.A.O.; methodology, M.B. and O.N.; validation, B.G.J.Y. and O.N.; formal analysis and investigation, L.S.B.A.I., M.B. and B.W.G.; resources, B.G.J.Y.; writing—original draft preparation, L.S.B.A.I.; writing—review and editing, H.Z.-D. and B.C.S.; supervision, R.S.; project administration, E.K. All authors have read and agreed to the published version of the manuscript.

**Funding:** This research received no external funding.

**Institutional Review Board Statement:** Not applicable.

**Informed Consent Statement:** Not applicable.

**Data Availability Statement:** No new data were created or analyzed in this study.

**Conflicts of Interest:** The authors declare no conflict of interest.

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
