# Peer review of "Analysis of Depigmenting Substances of Interest (Hydroquinone, Kojic Acid, and Clobetasol Propionate) Contained in Lightening Cosmetic Products Marketed in Burkina Faso"

_cosmetics, doi:10.3390/cosmetics10060154_

Round 1

Reviewer 1 Report

Comments and Suggestions for Authors

The manuscript details the presence of three depigmenting compounds in several cosmetic products.

The analytical data obtained shows that,  in a large part of the products analyzed the presence and the amounts of the three different compounds are not responding to the information declared on the label. The results could be useful to better regulate this typology of cosmetics.

I think that, on the basis of the results obtained, the discussion and the conclusions could be improved to increase the value of the manuscript.

On the other hand, I suggest in the future support the information that you have had at the analytical level with other data such as the response of specific cell lines (in vitro analysis) with the aim of evaluating the potential damage action that this kind of cosmetics could produce on the skin tissue.

Author Response

We thank the reviewers for reviewing the submitted paper. We agree with the reviewer comments and corrections are made in the revised manuscript.

Reviewer 1

The manuscript details the presence of three depigmenting compounds in several cosmetic products.

The analytical data obtained shows that, in a large part of the products analyzed the presence and the amounts of the three different compounds are not responding to the information declared on the label. The results could be useful to better regulate this typology of cosmetics.

We would like to thank the reviewer for this comment and the appreciations made.

I think that, on the basis of the results obtained, the discussion and the conclusions could be improved to increase the value of the manuscript.

As requested by the reviewer, corrections have been made in the revised manuscript. Additional comments were made to the discussion and conclusion (see last three paragraphs of the discussion section and second paragraph of the conclusion).

On the other hand, I suggest in the future support the information that you have had at the analytical level with other data such as the response of specific cell lines (in vitro analysis) with the aim of evaluating the potential damage action that this kind of cosmetics could produce on the skin tissue.

We agree with the reviewer suggestion.

This suggestion is take into account in the second stage of our study. This perspective was added at the conclusion level (see second paragraph of the conclusion).

We thank the reviewer for this suggestion.

Reviewer 2 Report

Comments and Suggestions for Authors

The submitted manuscript describes the determination of three depigmenting substances in lightening cosmetic products, using HPLC. The manuscript is clear and relevant for the journal. The work is presented in a well-structured manner. I suggest the authors to address the following minor revisions;

The authors used validated methods to determine the analytes. In section 2.2.5, the extraction should be elaborated (g of sample, solvent, etc.) and the preparation of the reference solution as well. Also, the limits of detection of the methods should be referred.

The manuscript contains five tables and three figures. Even though they properly show the data, some of them could be transferred in the supplementary material.

Table 5; check the product code (E1, E2, etc.). Are they similar with S1, S2, etc. in table 1?

Author Response

The submitted manuscript describes the determination of three depigmenting substances in lightening cosmetic products, using HPLC. The manuscript is clear and relevant for the journal. The work is presented in a well-structured manner.

We would like to thank the reviewer for this comment and the appreciations made.

I suggest the authors to address the following minor revisions;

The authors used validated methods to determine the analytes. In section 2.2.5, the extraction should be elaborated (g of sample, solvent, etc.) and the preparation of the reference solution as well. Also, the limits of detection of the methods should be referred.

In accordance with the reviewer’s suggestion, the section 2.2.5 was better developed and the relevant description has been made in the revised manuscript.

The manuscript contains five tables and three figures. Even though they properly show the data, some of them could be transferred in the supplementary material.

As requested by the reviewer, table 5 was transferred in the supplementary data.

Table 5; check the product code (E1, E2, etc.). Are they similar with S1, S2, etc. in table 1?

We agree with the reviewer comments. The product codes in table 5 have been corrected and harmonized with those in Table 1 (S1, S2, etc).